# Inequity in Access and Delivery of Virtual Care Interventions: A Scoping Review

**DOI:** 10.3390/ijerph19159411

**Published:** 2022-08-01

**Authors:** Sabuj Kanti Mistry, Miranda Shaw, Freya Raffan, George Johnson, Katelyn Perren, Saito Shoko, Ben Harris-Roxas, Fiona Haigh

**Affiliations:** 1Centre for Primary Health Care and Equity, University of New South Wales, Sydney 2052, Australia; 2Department of Public Health, Daffodil International University, Dhaka 1207, Bangladesh; 3RPA Virtual Hospital, Sydney 2050, Australia; miranda.shaw@health.nsw.gov.au (M.S.); freya.raffan@health.nsw.gov.au (F.R.); 4Sydney Institute for Women, Children and their Families, Sydney Local Health District, Sydney 2050, Australia; george.johnson@health.nsw.gov.au (G.J.); katelyn.perren@health.nsw.gov.au (K.P.); 5Health Equity Research Development Unit (HERDU), Centre for Primary Health Care & Equity, The University of New South Wales, Sydney Local Health District, Sydney 2050, Australia; s.saito@unsw.edu.au (S.S.); f.haigh@unsw.edu.au (F.H.); 6School of Population Health, University of New South Wales, Sydney 2052, Australia; b.harris-roxas@unsw.edu.au

**Keywords:** inequality, health equity, health services, virtual care, COVID-19, scoping review

## Abstract

The objectives of this review were to map and summarize the existing evidence from a global perspective about inequity in access and delivery of virtual care interventions and to identify strategies that may be adopted by virtual care services to address these inequities. We searched *MEDLINE*, *EMBASE*, and *CINAHL* using both medical subject headings (MeSH) and free-text keywords for empirical studies exploring inequity in ambulatory services offered virtually. Forty-one studies were included, most of them cross-sectional in design. Included studies were extracted using a customized extraction tool, and descriptive analysis was performed. The review identified widespread differences in accessing and using virtual care interventions among cultural and ethnic minorities, older people, socioeconomically disadvantaged groups, people with limited digital and/or health literacy, and those with limited access to digital devices and good connectivity. Potential solutions addressing these barriers identified in the review included having digitally literate caregivers present during virtual care appointments, conducting virtual care appointments in culturally sensitive manner, and having a focus on enhancing patients’ digital literacy. We identified evidence-based practices for virtual care interventions to ensure equity in access and delivery for their virtual care patients.

## 1. Introduction

Health inequities are referred to as those differences in health that are systemic, avoidable, unfair, and unjust [1]. A health equity approach recognises that not everyone has the same level of health or level of resources to address their health problems, and it may therefore be important apply different approaches in order to achieve similar health outcomes [2,3]. Health inequities are associated with a range of factors including age, gender, ethnicity, geographic location, and socioeconomic status [1,4]. A recent report has documented nine drivers of health inequity in relation to healthcare services: housing, income and wealth, health system and services, education, employment, social environment, transport, public safety, and physical environment [5]. Outcomes are determined by the dynamic interaction between service users and the health systems [6]. The World Health Organization (WHO) also identified gender, education, income, employment status, and ethnicity as the major factors associated with health inequity [7]. Health inequities are an established global phenomenon [8,9] and is a particular concern in a multicultural country with a history of settler colonialism, such as Australia [10,11]. Evidence demonstrates that the determinants of health inequity often lead to adverse health outcomes in the form of morbidities and mortality among vulnerable and marginalised populations [4,12,13].

Virtual care can be defined as “any interaction between patients and/or members of their care team occurring remotely, using technology with the aim of facilitating or maximising the quality and effectiveness of patient care” [14]. It has been identified as an approach that may partially address health inequities through improving access and availability of health services [15,16]. However, there are also concerns that virtual care services could exacerbate existing health inequities if services are not accessible, available, and acceptable to vulnerable population [17,18]. Virtual care interventions received particular attention during the COVID-19 pandemic, as many health services rapidly transitioned to providing virtual care services as an emergency method of reaching their clients [19]. The restriction of in-person health services and the rapid implementation of virtual care has been driven by necessity but also presents a significant opportunity to develop and strengthen the provision of virtual care [20,21,22]. However, this expansion in virtual care services using healthcare technologies also created the potential for the widespread digital divide to act as a potent barrier in successful implementation of virtual care interventions and a cause of health inequities. The digital divide is defined as disproportionate access and utilization of health technology and internet among certain population groups, characterised by their geographical, social, and geopolitical criteria or other features [23]. There are suggestions of a “digital paradox” where the “population groups that could potentially benefit most from digital innovations are the ones that would experience the highest barriers to access” [24].

Recently, several studies were conducted on the expansion of virtual care interventions, particularly in relation to the COVID-19 pandemic [25,26,27]. Many of these studies considered virtual care as a way of minimising the risk of COVID-19 transmission [26,28], to triage during emergency responses [21] and monitoring patients within their homes [21]. One such intervention is the RPA Virtual Hospital (rpavirtual), launched in February 2020 as a new model of care that combines integrated hospital and community care with digital solutions. It was the first service to introduce virtual care for COVID-19-stable patients in isolation in New South Wales, Australia, and has been demonstrated to be widely accepted by patients [29]. However, the potential equity issues related to rpavirtual and other similar virtual care interventions have not yet been adequately explored and described. Therefore, this scoping review aims to map and summarize the knowledge about equity issues in the access and delivery of virtual care interventions and to identify strategies to address potential inequities that may be adopted by virtual care services.

## 2. Materials and Methods

This scoping review is reported following the guidelines of PRISMA-ScR (Preferred Reporting Items for Systematic Reviews and Meta-analysis extension for Scoping Reviews) [30]. The review protocol is registered at the website of Centre for Primary Health Care and Equity, UNSW Sydney (https://cphce.unsw.edu.au/research/rapid-literature-review-identify-equity-issues-access-and-delivery-virtual-care, accessed on 1 March 2022), and PRISMA-ScR is provided in Appendix A.

### 2.1. Data Sources

We searched for peer-reviewed articles in electronic databases: *Medline*, *EMBASE*, and *CINAHL*. Both medical subject headings (MeSH) and free-text keywords were used to search relevant articles in these databases that were published in the English language between January 2010 and January 2021. The detailed search strategy is presented in Table 1.

### 2.2. Study Selection

The articles yielded in the initial database searches were assessed by two independent reviewers in relation to the inclusion and exclusion criteria developed for this study (Box 1). All of the steps of study selection procedure were performed in Covidence (https://www.covidence.org, accessed on 1 March 2022). In the first stage, the title and abstract of the articles were assessed by two reviewers. The articles that passed this initial screening stage entered full text screening. The full texts of these articles were obtained, and more in-depth assessment was carried out against the inclusion and exclusion criteria. The reason for the exclusion for each of the articles was also noted at this stage. Any difference in assessment between the reviewers was resolved by discussion.
Box 1Inclusion and exclusion criteria.**Inclusion criteria**Published in EnglishPublished between January 2010 and January 2021Studies exploring equity in ambulatory services offered virtuallyCarried out in OECD countriesEmpirical studies**Exclusion criteria**Published in language other than EnglishPublished before January 2010Studies not exploring equity in ambulatory services offered virtuallyStudies exploring robotic/tele-surgeryStudies carried outside OECD countriesCommentary/review/opinion pieces


### 2.3. Data Extraction

The data were extracted from the included studies in a Microsoft Excel template developed by the authors. Information including country, study setting, study design, study participants, characteristics of the intervention/study, type of virtual care modalities, type of inequity issues identified/addressed, main findings, summary of the results, and relevance to virtual care interventions were extracted.

### 2.4. Data Mapping

As the objective of scoping reviews is to map and summarize the available evidence, we performed descriptive analysis, which involved frequency counting and basic thematic coding [31].

## 3. Results

### 3.1. Search Results

Database searches yielded a total of 3021 articles, from which 1990 underwent screening after removal of the duplicates. The assessment of the title and abstract of the articles resulted in the exclusion of 1901 articles, and 89 articles underwent full-text screening. Finally, 41 articles satisfied the selection criteria and were included in the review (Figure 1). The detailed characteristics of the included studies are presented in Appendix A.

### 3.2. Study Settings

Of the forty-one included studies, thirty-one were conducted in the USA, three were carried out in Australia [32,33,34], two in Canada [35,36], one in Italy [37], one in China [38], one in Germany [39], one in Norway [40], and one in Scotland [41]. The studies were carried out either in a community or in a clinical setting, such as a hospital or primary care.

### 3.3. Study Designs

A range of study designs were used in the included studies. Twenty-three of the included studies followed a cross sectional design [5,32,34,36,37,38,39,40,41,42,43,44,45,46,47,48,49,50,51,52,53,54,55], five studies carried out retrospective analysis of the collected data [56,57,58,59,60], six studies followed cohort design [61,62,63,64,65,66], two were randomised controlled trials [67,68], and two followed a mixed-method design [33,69]. One study followed a combination of retrospective analysis and cross-sectional study design, [70] while the study design was not clear in two studies [35,71].

### 3.4. Type of Participants

The participants in most of the studies were adults, often with chronic conditions such as diabetes [45], cardiovascular disease [39], and mental health problems [61]. Most of the studies considered both native English speakers and those speaking languages other than English. Only a few studies considered all participants speaking a language other than English, such as Spanish [56] or Chinese [34]. Several studies examined outcomes of specific cultural and ethnic minorities. However, since most of the included studies were conducted in the USA, the population groups were mostly Black, Hispanic, and African American [46,48,50,54,55,60,66,67,68].

### 3.5. Virtual Care Modalities

The included studies considered several modalities of virtual care interventions ranging from video conferencing [37,41,42,44,46,48,49,51,57,59,60,61,62,63,64,66,67,71], teleconferencing [34,35,46,48,49,51,53,54,56,57,58,60,62,63,66,71], messaging [42,45,50], emails [42], health apps [5,39,40,50], patient portals [58,61,68,70], personal health records [59,61], and eHealth service use via the Internet [32,40,47,69].

The majority of the studies described the use of virtual visits (either video or audio) in comparison to face-to-face visit or video visit in comparison to audio/tele visit. Several video conferencing platforms, such as Zoom (https://zoom.us/, accessed on 1 March 2022) or Microsoft Teams (https://www.microsoft.com/en-au/microsoft-teams/group-chat-software/, accessed on 1 March 2022), were used to perform video visits in the reported studies. Some of the studies reported on non-synchronous communication tools such as text messaging, health apps, patient portals, or eHealth service use. Text messaging, health apps, or patient portals were generally used to book appointments with service providers, access health information, track health outcomes, or communicate with health service providers. On the other hand, eHealth services were offered to promote online learning, counselling, and information sharing, and these aims were accomplished through browsing search engines, health apps, social media, and video services.

### 3.6. Types of Inequity Issues Identified/Addressed

#### 3.6.1. Cultural and Ethnic Inequities

Twenty-one studies [33,42,44,45,46,48,50,54,55,57,59,60,61,62,63,64,65,66,67,68,70] explored cultural and ethnic inequities in access to virtual care services and outcomes. The majority of these found that cultural and ethnic minorities, including those of African American, Black, Hispanic or Latinos, Asian American, Aboriginal and Torres Strait Islander, or Filipino descent, were less likely to access virtual care services compared to the White participants. For example, in their study, Schifeling and colleagues [60] found that non-White patients were less likely to have a video visit than White patients. Likewise, Walker et al. [68] found that African American patients used the patient portal less than White patients (40.4% difference, *p* = 0.004). However, four studies [42,44,50,65] reported a different result where the likelihood of using virtual care services was higher among the cultural and ethnic minorities compared to White participants.

#### 3.6.2. Sociodemographic and Socio-Economic Inequities

Older people were identified as experiencing significant barriers to accessing and using virtual care services in most of the studies [5,32,36,38,39,41,43,45,46,47,48,49,50,55,60,61,63,65,68,70,71]. For example, Leng et al. [41] found that the patients under 60 years were over two times more likely to use video consulting (odds ratio (OR) 2.2, 95% CI 2.1–6.6). Nelson et al. [45] also pointed out that the probability of responding to texts tended to increase from about age 25 years until roughly age 50 years and then appeared to decrease with increasing age. Eberly et al. [64] further noted that younger participants were more likely to be engaged with video call appointments compared to telephone call. The only exception was reported by Pierce et al. [46], where age of 65 years and above was associated with higher odds of virtual care use (OR 1.21, 95% CI 1.05–1.40). It is also notable to mention that all nine studies [33,39,46,53,57,61,63,64,65] that explored the role of gender in accessing virtual care services found that females were less likely to use virtual care services compared to males. Two studies [63,65] also found that unmarried participants were less likely to access virtual care services. Meanwhile, Wegerman et al. [66] found that participants who were single or previously married (separated, divorced, widowed) had higher odds of completing a telephone appointment, while married participants were more likely to complete a video appointment.

Thirteen studies explored the use of virtual care in relation to the socioeconomic status of the participants, and all of these found that lower socioeconomic status was associated with lower use of virtual care services [5,32,33,38,39,40,47,48,50,51,61,63,67]. Alam et al. [32] reported that access to virtual care services was lower among participants from disadvantaged socioeconomic backgrounds. Likewise, other studies [5,33,38,40,48,50,51,61,63] also reported that low socioeconomic status was associated with decreased access to virtual care services. Not surprisingly, some of the included studies that explored the role of education in accessing virtual care services [5,32,33,38,39,40,47,48,50,67] also found that participants with lower education status were less likely to access the virtual care services.

#### 3.6.3. Inequity Issues Related to Digital/eHealth Literacy

Seven studies [32,38,39,41,45,56,69] reported a lack of digital/eHealth literacy among the participants as a significant barrier to accessing virtual care services. Ernsting and colleagues [39] found that mHealth app users had higher levels of eHealth literacy compared to non-app users. A study [69] also reported that eHealth literacy increase was associated with a 3% increase in the number of searches for health information on the internet (beta = 0.03, 95% CI 0.00–0.06). Meanwhile, Leng et al. [41] found that higher computer proficiency correlated with an increased willingness to engage in video consultations.

#### 3.6.4. Technological Inequities

Several studies [32,37,48,50] also found that improved access to digital devices and internet can increase the use of virtual care services. Arighi et al. [37] reported that issues such as a lack of devices (computers, phones, or tablets) with internet connection and poor internet connections were the main causes of failed virtual care. Alam et al. [32] pointed out that access to broadband internet services was associated with increased use of virtual care services.

## 4. Discussion

This review was conducted to explore inequity issues in relation to access to and delivery of virtual care interventions and to consider the international evidence of actions to address inequity issues that may be adopted by or provide learnings for rpavirtual and other similar virtual care interventions. The main drivers of inequity in access to virtual care identified in the literature review were relatively older age, unemployment, less income, lower education level, belonging to cultural or ethnic minorities, lack of access to digital devices or good internet connection, and lack of digital/eHealth literacy.

In recent times, due to the COVID-19 pandemic, virtual care interventions have been widely used due to restricted in-person health service delivery [25,26]. It has also been documented that the patient experience and their acceptance of virtual care during this pandemic has been generally good [72,73]. At the same time, it is also worth noting that the expansion of this digital innovation without due consideration of strategies to address inequity of access has the potential to increase health inequities due to poverty, digital health literacy, and lack of access to digital technology among some of the population [74].

Reviews carried out during the COVID-19 pandemic [75,76] also stressed the importance of virtual care interventions as an alternative to in-person health service delivery during a period of restrictions on face-to-face health service delivery. Doraiswami et al. (2020) [75] reported that virtual care could play a pivotal role in the health sector in future, but its feasibility and implementation in a resource-poor setting is challenging. In this regard, it is critical to mention that future virtual care interventions will be influenced by broader health and clinical governance agendas and directions in investment across systems.

While some recent reviews [77,78,79] have highlighted the effectiveness of virtual care as a way of delivering health care in a cost-effective way, with improved patient communication, outcomes, and satisfaction, the equity dimension of the virtual care interventions is not fully addressed in these reviews. The present review has helped to bridge the knowledge gap around inequity issues associated with virtual care and identified areas for further research.

The present review highlighted that access to virtual care services is particularly limited among patients from ethnic minorities, which suggests there is a need to carefully tailor services to ensure equitable access. Multilingual and culturally sensitive virtual care services can be of high value in this regard. For example, a culturally sensitive approach documented by Shaw et al. (2013) [34] could be to address cultural diversity in the developing of a virtual care intervention. This qualitative study was conducted among Chinese and Arabic patients and their carers to explore their willingness to take part in a telephone-based supportive-care intervention. The majority of the study participants supported the provision of a culturally sensitive intervention in their own language via an online platform. However, the participants identified that confidentiality of the clinical information was a concern and preferred an initial in-person appointment with patients to increase participation. It was also suggested that there should be the provision of an “on-call” support process initiated by patients to provide patients with access to assistance in times of high need between scheduled calls.

Access to virtual care services is linked to the level of digital literacy of the patients. For example, Ernsting et al. [39] and Guendelman et al. [69] strongly emphasised the importance of improving digital literacy of patients in order to address inequity of access to virtual care services. Older people and individuals with limited digital health literacy are less likely to access virtual care services and require targeted support. The present review indicates that availability of younger caregivers or caregivers with higher digital literacy can result in increased access to virtual care services [37].

Consideration of different levels of digital and health literacy across patients should be a part of routine planning for virtual care services. For example, an educational component can be incorporated in interventions to increase virtual care literacy among vulnerable patients. In addition, delivery methods can be updated, for example, by adapting portals to be comfortably used by less digitally literate patients or appropriately tailoring information or platforms to vulnerable patients.

Virtual care service delivery planning should consider the variances in service uptake between different socioeconomic classes. Access to digital resources influences a person’s capacity to access and utilise virtual care. Research has also documented that the digital divide in terms of access to digital devices and strong internet connectivity is significant among people with lower level of education and lesser income [80,81]. When engaging patients with virtual care services, consideration should be given as to whether patients have access to appropriate devices and a reliable internet connection. rpavirtual and other similar virtual care interventions should include in their referral process that patients require devices and internet connection to access services.

This review was subject to some limitations. There are several synonyms used to represent inequity issues in the literature. While we were broad in searching the literature, we may still have missed some articles utilising different terminology. However, we explored both the MeSH terms and keywords to address this. We also limited our searches to three major databases, and there could be additional relevant articles available in other databases. We searched for only the peer-reviewed articles and therefore might have missed some grey publications.

We restricted our searches to English literature only, and therefore, we could miss relevant articles that are written in a language other than English. Furthermore, we only searched for studies published in last decade (January 2010–January 2021); therefore, we could miss some articles published before 2010 and after 2021.

## 5. Conclusions

This review highlights that while there is potential for virtual care to improve health service delivery, particularly during the COVID-19 pandemic, there can be widespread inequities in access to and delivery of virtual care interventions. These inequities are based on sociodemographic characteristics of the participants, such as age, gender, and ethnicity as well as other factors, such as access to appropriate digital technology, digital and health literacy, cultural acceptability, and trust and perceived quality of care. This review has identified several promising practices, such as the inclusion of young and educated caregivers, providing culturally sensitive interventions, and improving digital health literacy among patients. These strategies can be adopted by rpavirtual and other virtual care interventions to ensure equity in access and delivery of virtual care services. Future research should focus on how these promising practices can be implemented in clinical settings.

## Figures and Tables

**Figure 1 ijerph-19-09411-f001:**
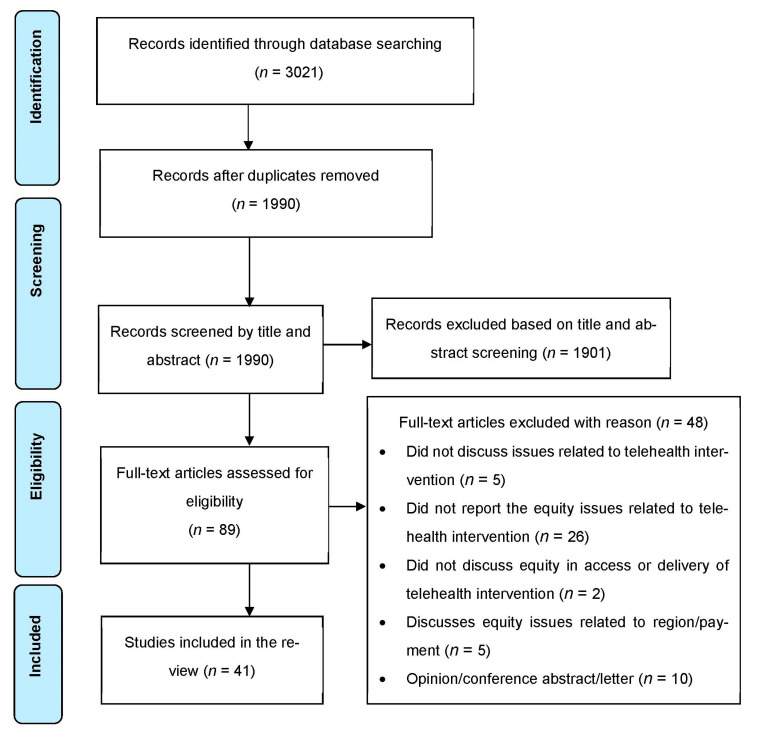
PRISMA diagram of study selection.

**Table 1 ijerph-19-09411-t001:** Search strategy.

Sl.	Search Terms
1	“telemedicine” [MeSH Terms] OR “telemedicine” [Text Word]
2	“tele medicine” [Text Word]
3	“telehealth” [Text Word]
4	“tele health” [Text Word]
5	“tele-health” [Text Word]
6	“e-health” [Text Word]
7	“teletherapy” [Text Word]
8	“virtual care” [Text Word]
9	“virtual health” [Text Word]
10	1 or 2 or 3 or 4 or 5 or 6 or 7 or 8 or 9
11	“disparit*” [Text Word]
12	“health equity” [MeSH Terms] OR “health equity” [Text Word]
13	“equit*” [Text Word]
14	“inequit*” [Text Word]
15	“inequalit*” [Text Word]
16	“healthcare disparities” [MeSH Terms] OR “health care disparities” [Text Word]
17	“health status disparities” [MeSH Terms] OR “health status disparities” [Text Word]
18	10 or 11 or 12 or 13 or 14 or 15 or 16 or 17
19	10 and 18

## Data Availability

Not applicable.

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
