# Peer review of "Inequity in Access and Delivery of Virtual Care Interventions: A Scoping Review"

_ijerph, 2022, doi:10.3390/ijerph19159411_

Round 1

Reviewer 1 Report

Dear Authors,

thank you for the interesting paper. Below you will find my recommendations.

In the abstract, the total number of studies found should be added.  

1. Introduction

This section lacks a presentation of theoretical and empirical findings on the drivers of health inequity in the context of health care (e.g., access) as well as the role of existing health care inequity on health inequity in terms of morbidity and mortality. Also missing is reference to relevant models of the digital divide in health care. The section on the rpavirtual project could be significantly shortened.

2.2 Data sources

Furthermore, the methods section should justify why studies from 2010 onwards are considered, as an enormous temporal influence can be assumed in this field. This should also be mentioned in the limitations.

2.6 Narrative Analysis

The methodological approach should be described. Considering the decision for a scoping review and the corresponding methodological approach, the reference to meta-analysis is not meaningful.

Table 2

The contents, especially for main findings, should be presented more concisely so that readability is increased. The detailed table could be provided as a supplement. Formatting should be changed, not centered.

Authors should consider using different or no numbering when labeling studies, as this may cause confusion with numbering in citations. In addition, it would be useful to add the citation number with the authors.

Table 3

Also here it would be worth considering whether this should be provided as a supplement and at the same time the text should be expanded somewhat to include the core results from the table, e.g. which studies have a moderate quality assessment and what the main reasons for this are.

3.5 Virtual care modalities

Here it would be interesting if additional information on the individual virtual care modalities could be given. As the table shows, there are variances that should be outlined in this section.

3.6 Types of equity issues identified/addressed

In this section, a differentiation of the results with respect to the drivers of health inequities in health care would be interesting. Specific indications may already be available here in the underlying studies.

3.6.3 Socio-economic inequalities

This heading should be adapted to: sociodemographic and socio-economic inequalities.

4 Discussion

Lines 91-95: Reasons are given here that are not highlighted in the results. These aspects should already be taken up in the results as described.

More structure should be given to the discussion; the differences found on the one hand and the possible causes on the other hand could be a good starting point. Also, the prioritization - e.g., it is reported in great detail on health literacy - should be reviewed. This should be as consistent as possible with the recommended supplements in the introduction.

Reviewer 2 Report

Thank you for the opportunity to read and review this interesting article. This is an important topic and health equity is an important meeting fields. While I feel it is an important area, I feel the current methods are not sound or trustworthy and these must adhere to the standard reporting guidelines for scoping reviews and also there needs to a higher quality approach to the field of health equity- I refer authors to the work of the Campbell and Cochrane Colaborations Equity Methods Group.

I believe the title should state more clearly health inequity or health equity issues from the start.  This paper essentially attempts to apply an equity lens to an exploratory search of the literature, but it fails to incorporate and follow high quality reporting guidelines for scoping reviews, uses a very unsatisfactory appraisal approach that nefariously concludes the papers are all good quality, please note- most scoping review reporting guidelines recommend not attempting appraisals because of the difficulty doing this consistently and rigoursly across a range of study types- I would recommend no appraisal is better than their current superficial one. This paper also starts to feel like a systematic review of outcomes and this is dangerous territory for an exploratory scoping review. This scoping review need to be more modest in in interpretations and conclusions.

The topic is interesting but the abstract is not well written, the methods and appraisals not sound and so a considerable re-write is needed.

Round 2

Reviewer 2 Report

The authors have done a large and broad review. The topic is of interest but not very well focused.  

I detect several areas still in need for work.  For example, the intro begins with a strange definition of health equity and does not provide a references. 

The next sentence does provide a more conventional definition of health inequity but the reference in the reference list is not complete. 

The authors have change some references to health equalities to health inequities, but not all. There a many poorly writing and difficult to read sentences and typo areas, for example - zoom.  Sometimes the authors talk about synthesizing results (but I'm not sure how) and then sometimes they talk about mapping. Consistency of precision is needed. 

This is a large scoping review that would need much more work to be ready for publication.  
